# On the Use of the Digital Twin Concept for the Structural Integrity Protection of Architectural Heritage

Annalaura Vuoto [1,*], Marco Francesco Funari [2] and Paulo B. Lourenço [1]

1   ISISE, Department of Civil Engineering, University of Minho Campus de Azurém,
    4800-058 Guimarães, Portugal; pbl@civil.uminho.pt
2   School of Sustainability, Civil and Environmental Engineering, University of Surrey, Guildford GU2 7XH, UK;
    m.funari@surrey.ac.uk
*   Correspondence: annalauravuoto1307@gmail.com

**Abstract:** Undoubtedly, heritage buildings serve as essential embodiments of the cultural richness and diversity of the world's states, and their conservation is of the utmost importance. Specifically, the protection of the structural integrity of these buildings is highly relevant not only because of the buildings themselves but also because they often contain precious artworks, such as sculptures, paintings, and frescoes. When a disaster causes damage to heritage buildings, these artworks will likely be damaged, resulting in the loss of historical and artistic materials and an intangible loss of memory and identity for people. To preserve heritage buildings, state-of-the-art recommendations inspired by the Venice Charter of 1964 suggest real-time monitoring of the progressive damage of existing structures, avoiding massive interventions, and providing immediate action in the case of a disaster. The most up-to-date digital information and analysis technologies, such as digital twins, can be employed to fulfil this approach. The implementation of the digital twin paradigm can be crucial in developing a preventive approach for built cultural heritage conservation, considering its key features of continuous data exchange with the physical system and predictive analysis. This paper presents a comprehensive overview of the digital twin concept in the architecture, engineering, construction, and operation (AECO) domain. It also critically discusses some applications within the context of preserving the structural integrity of architectural heritage, with a particular emphasis on masonry structures. Finally, a prototype of the digital twin paradigm for the preservation of heritage buildings' structural integrity is proposed.

**Keywords:** digital twin; built cultural heritage; structural integrity; masonry structures; preservation

## 1. Introduction

Digital transformation is spreading in several domains, ranging from the industrial production of mechanical components to infrastructure monitoring and maintenance. Nevertheless, the architecture, engineering, construction, and operation (AECO) industry is one of the most reluctant industries to introduce a comprehensive digitalisation process [1]. **Figure 1** reports a helpful classification of the leading digital technologies currently adopted in the construction sector performed by the European Construction Sector Observatory (ESCO) [1]. The technologies used in the digitalisation process can be categorised into three groups: data acquisition, process automation, and digital information and analysis. Data acquisition has become a relatively simple task thanks to the rapid development of related technologies. However, the actual use of digital data represents the future of digitalisation, and should be performed by employing state-of-the-art automated processes. Furthermore, by automating certain activities, the final quality of the project improves, workers are less exposed to risks, and new materials and techniques can be deployed. In this framework, the digital twin (DT) concept is intended as the highest expression of the digital information and analysis management processes.

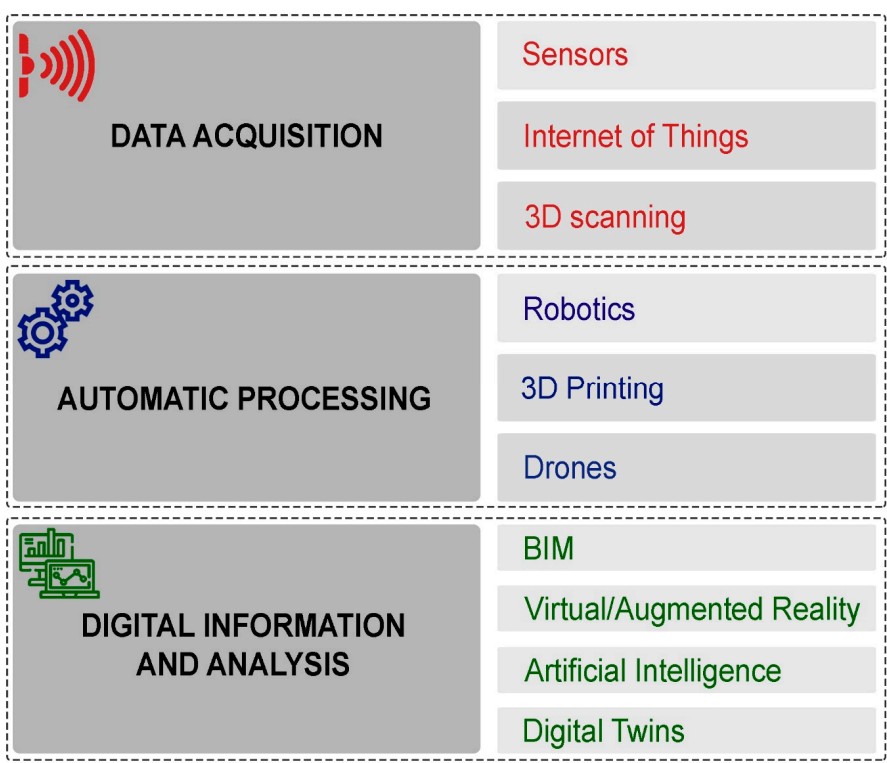

**Figure 1.** The three categories of digital technologies in construction. Redrawn from the classification provided in [1].

The DT concept is often misunderstood as a basic geometric digital representation of an asset, which overlaps with computer-aided design (CAD) and building information modelling (BIM). The connection between the physical asset and its digital counterpart marks its difference from any other digital model or replica. Such a connection gathers dynamic data, such as forces and loads, temperature variations, and rate-dependent phenomena [2]. Therefore, while CAD models are static representations of shapes, a DT should be seen primarily as a computational model that can perform analyses and predictions related to the behaviour of physical assets under changing conditions rather than as a simple geometric representation [3]. Understanding this concept is crucial for utilising a DT effectively. Depending on its maturity, a DT can range from a simple 2D or 3D model with a basic level of detail all the way to a fully integrated and highly accurate model of an entire asset. There is no single solution or platform used to provide a DT as it is to be intended as a paradigm rather than a technology. The DT is a concept of leveraging knowledge on an asset by managing and interpreting huge data sets [4]. Hence, in this work, the terms "paradigm" and "concept" refer to the whole process, while the term "model" only refers to the digital counterpart or replica. A general model [5] sets three levels of DT maturity, namely, (i) partial: a partial DT contains a small number of data sources, it is useful for monitoring a key metric or state, and it enables the quick development of device-to-platform functionality and contains enough data sources to create derivative data; (ii) clone: a clone DT contains all meaningful and measurable data sources from an asset, and it is useful in the prototyping and data characterisation phases of IoT development; and (iii) augmented: an augmented DT enhances the data from the connected asset with derivative data, correlated data from federated sources, and/or intelligence data from analytics and algorithms. In addition, technical experts within the built environment domain developed a maturity model tailored for the specific application area. The model [4] comprises the following five levels of maturity, described according to their defining principle and to be interpreted on a logarithmic scale of complexity and connectedness: (0) reality capture (e.g., point cloud, drones, photogrammetry, or drawings/sketches); (1) 2D map/system or 3D model

(e.g., object based, with no metadata or BIM); (2) connect model to persistent (static) data, metadata, and BIM Stage 2 (e.g., documents, drawings, and asset management systems); (3) enrich with real-time data (e.g., from IoT sensors); (4) two-way data integration and interaction; and (5) autonomous operations and maintenance.

However, while a fully developed DT remains an objective, research and industry are delivering partial pieces of this concept rather than a complete paradigm's implementation [4].

Despite researchers' significant efforts to demonstrate the potential of DTs in solving industry problems, their application in the civil engineering field is still limited to a few pilot projects [1]. However, researchers and practitioners agree on the potential key role that implementing the DT paradigm correctly in the AECO domain could play, particularly in enhancing the development of the sustainable management of built assets. In this context, it is widely acknowledged that the building industry is a significant contributor to carbon emissions, primarily through the energy consumption required for heating, cooling, and lighting buildings and the production and transportation of construction materials. This excessive carbon footprint contributes to serious environmental issues, such as climate change, air pollution, and the depletion of natural resources. Furthermore, it is projected that the world's population will continue to increase over the next few decades, which will result in a higher demand for housing and energy consumption. This demand will likely lead to more construction and infrastructure development, resulting in even more significant carbon emissions.

Extending the life of existing structures and infrastructures can significantly reduce carbon-consuming activities. Retrofitting buildings to improve energy efficiency and using sustainable materials in construction can help reduce carbon emissions and mitigate the building industry's impact on the environment. Therefore, it is crucial to prioritise the readaptation of existing structures and infrastructures as a sustainable solution to meet the growing demand for housing and energy consumption. Built cultural heritage (BCH) presents a unique opportunity to develop sustainable solutions that address the challenges posed by climate change, population growth, and urbanisation. BCH refers to buildings, landscapes, and infrastructure that have cultural, historical, and architectural significance and are valued by communities for their cultural and symbolic importance.

The Venice Charter [6], a set of guidelines developed by the International Council on Monuments and Sites [7] for the preservation and restoration of cultural heritage sites, emphasises the importance of preserving the authenticity and historical significance of heritage buildings and using compatible materials and techniques in restoration and rehabilitation works. Following the principles of the Venice Charter, the modern approach aims to prioritise the preservation of the structural integrity of heritage buildings through real-time monitoring systems and targeted interventions. State-of-the-art recommendations suggest the use of real-time monitoring systems to detect and track progressive damage to existing structures. These systems can provide early warning for potential structural failures, allowing for timely intervention to prevent catastrophic damage. Additionally, it is recommended to avoid massive interventions that can compromise the structural integrity of heritage buildings and instead use targeted interventions that address specific areas of concern. However, such an approach is not fully developed due to the limited deployment of dense sensor networks in heritage buildings and the underdeveloped potential of using up-to-date digital information and analysis technologies to preserve BCH.

## 2. AECO Domain: Is the DT Concept Fully Understood?

The implementation of the DT paradigm in the AECO domain is at the triggering stage. Errandonea et al. [8] analysed the distribution of publications on DT across various fields, taking into account 167 contributions collected from the "Scopus" and "Web of Science" databases. **Figure 2** shows that the AECO domain represents a small percentage of the reviewed publications.

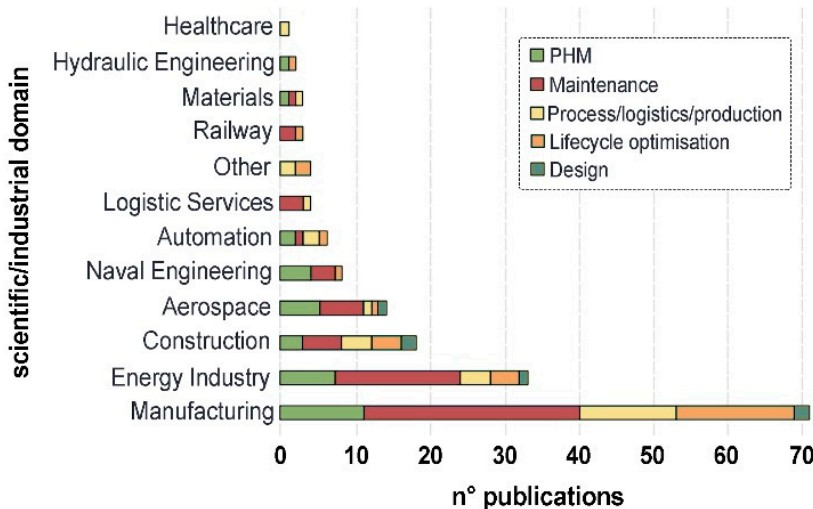

**Figure 2.** Publication distribution across various domains. Redrawn from Errandonea et al. [8].

Despite the growing interest and importance of DT in civil engineering, the available literature still falls short of providing adequate knowledge on the technology [9]. Unlike other fields, the implementation of DT in civil engineering has been influenced by the pre-existence and widespread adoption of the BIM paradigm, particularly in modern buildings' operational and maintenance (O&M) phase. However, there is a common mistake of overlapping DT and BIM technologies in the literature [10–13]. DT has become a popular buzzword in the construction industry, yet its underlying principles and applications are not always clearly understood. Some literature on the implementation of DT in civil engineering refers to it as a BIM model that is gradually incorporated throughout the building's life cycle, as illustrated in **Figure 3**. It is a fact that BIM is a digital representation of a building's physical and functional characteristics, helping manage its design, construction, and operation. Hence, users must recognise that DT goes beyond BIM. It involves creating a digital replica of a physical asset, such as a building or infrastructure, that uses real-time data and analytics to provide insight into its performance, maintenance needs, and potential issues. In other words, DT is a dynamic and interactive model that continuously learns from the physical asset and updates itself accordingly.

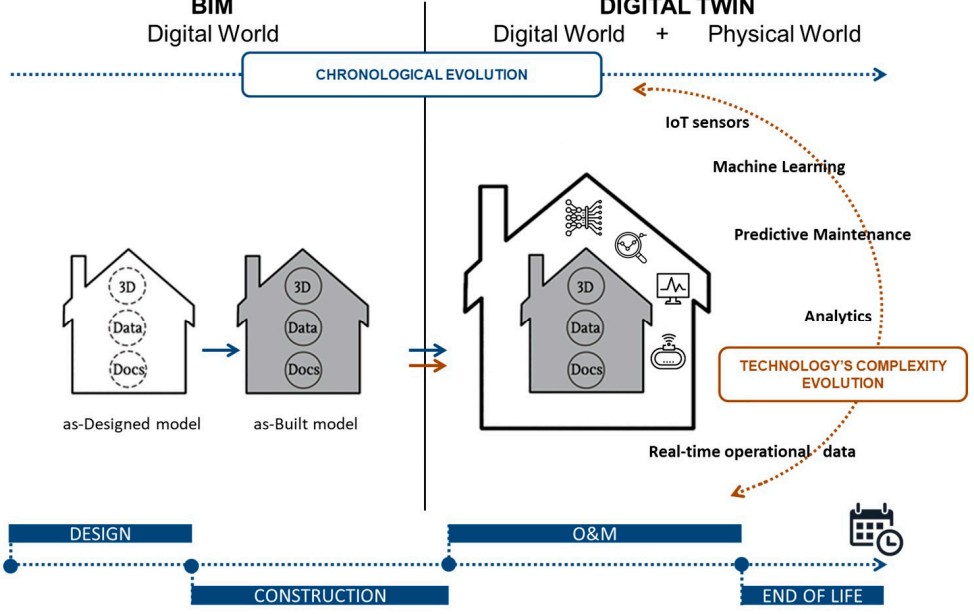

**Figure 3.** Digital twin (DT) as an evolution of building information modelling (BIM).

The lack of a clear definition for DT rises from a search of the literature. Conducting rigorous research is crucial to advancing the DT concept in civil engineering and defining its key features, applications, and limitations (see **Figure 3**).

Camposano et al. [9] reported opinions from prominent representatives of the AECO industry within the scope to conceptualise the DT paradigm. DT was identified as a digital model (CAD or BIM). Such a visual representation is enhanced with real-time data to monitor the asset state by integrating statistics or artificial intelligence (AI) models to enable intelligent predictive capabilities. DT could ideally replace human intervention and autonomously determine action on behalf of the physical asset. They concluded that CAD or BIM digital models are generally considered descriptive, whereas DT is expected to be more dynamic, representing geometry and structural behaviour evolution under specific boundary condition changes.

Deng et al. [11] proposed a five-level ladder taxonomy reflecting the evolution of the concept from BIM to DT. **Figure 4** represents their discretisation: Level 1—BIM, Level 2—BIM-supported simulations, Level 3—BIM integrated with IoT, Level 4—BIM integrated with AI techniques for predictions, and Level 5—ideal DTs. Each level is to be understood as an enhancement of the basic digitalisation technology (i.e., BIM) that allows for providing ever-increasing services and benefits. According to the authors, an ideal DT contains real-time monitoring and prediction capabilities and enables automated feedback control for the adjustment of building parameters when necessary. Deng et al. [11] also identified specific implementations of DTs in building projects based on both the area of application (i.e., research domain) and the level of complexity of the employed technology, according to the levels shown in **Figure 4**. As regards BIM models used as support for simulations (Level 2), they found implementations for (i) construction process simulations, (ii) energy performance evaluation, and (iii) thermal environment evaluation. In addition, applications for assessing embodied carbon, lighting, and acoustics were also referenced. Moreover, BIM models integrated with IoT techniques (Level 3) are used for (i) construction process monitoring; (ii) energy performance management; (iii) indoor environment monitoring; (iv) indoor thermal comfort; (v) space management; and (vi) hazards monitoring; (vii) community monitoring. When real-time predictions are added to BIM models and IoT sensors (Level 4), applications are found for (i) construction process prediction; (ii) thermal comfort prediction; (iii) management of indoor systems; (iv) life cycle cost (LCC) of building facilities; and (v) smart cities DT.

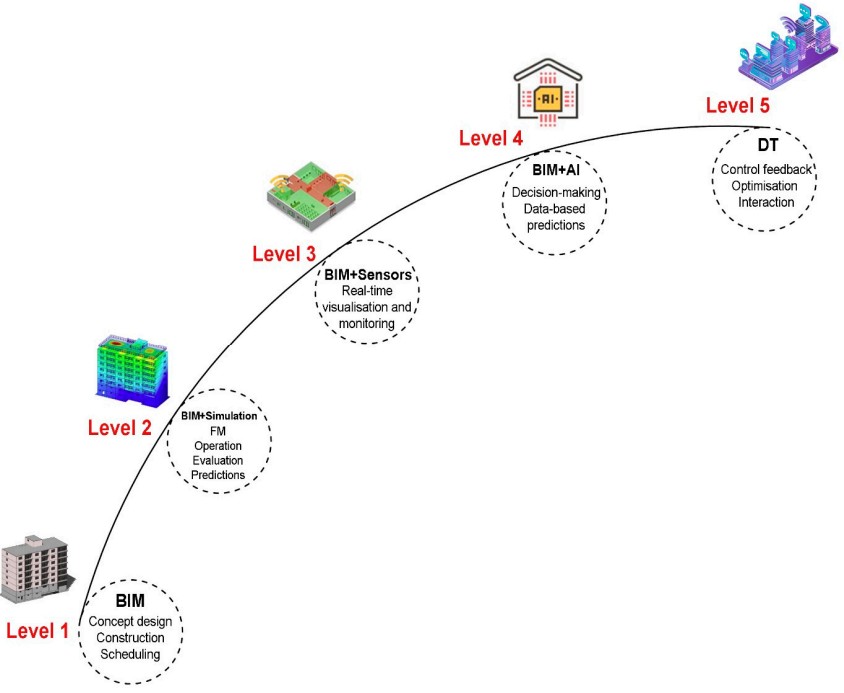

**Figure 4.** From BIM to DT. Redrawn from Deng et al. [11].

Other authors [14–17] correctly treated DT and BIM as independent concepts, underlining differences in terms of the final utility for the users. According to Jiang et al. [14], DT emphasises the existence of the physical counterpart, whereas a BIM model can be generated before the physical model is built. Khajavi et al. [15] listed three main differences between BIM and DT. Firstly, BIM was designed to improve design and construction efficiency, while DT is designed to monitor or simulate the behaviour of a physical asset, improve its operational efficiency, and enable predictive maintenance. Furthermore, while BIM is suitable for integrating cost estimation and time schedule data to enhance the efficiency of a construction project, DT is designed to integrate real-time sensor or data readings and data analysis tools resulting in a boosted interaction between the building and the environment and users.

Sacks et al. [16] stated that the DT should not be viewed simply as a logical progression from BIM or as an extension of BIM tools integrated with sensing and monitoring technologies. Instead, DT is a holistic model that prioritises closing the control loops by basing management decisions on reliable, accurate, thorough, and timeous information. A further consideration was made by the authors, who noted how BIM tools provide excellent product design representations using object-oriented vector graphics, which is not ideal for incorporating the raster graphics of point clouds acquired through the scanning of physical assets. This is particularly inconvenient for the implementation of the DT concept for existing buildings. In fact, an object-based approach is often not the most suitable for their representation, whereas a reality-based approach realised by remote sensor acquisitions could be more effective.

*Structural Integrity Protection of Architectural Heritage: Is the DT Paradigm Helping?*

The literature review shows that limited attempts have been made to implement the DT paradigm in the context of heritage buildings [3,18]. Most existing applications were designed for built infrastructures, and the few techniques that were tested in the heritage building domain were limited to facility management. When it comes to structural engineering, most of the studies focused on the structural monitoring and maintenance issues of infrastructure such as bridges. Few studies addressed the key features of the DT paradigm, such as real-time updating and learning of virtual models or automated decision making.

There are a few studies that propose a suitable framework for the implementation of the DT paradigm in structural engineering. Chiachío et al. [19] provided a functional DT technology integration framework within the structural O&M context, suitable for simulations, learning, and management.

Ramancha [20] proposed a structural digital twin for SHM and damage prognosis (DP) of large-scale civil infrastructure by implementing a Bayesian finite element (FE) model updating framework. However, in the context of BCH structural integrity preservation, no studies define a suitable framework.

The most recurrent applications of the DT paradigm in the field of BCH conservation found in the literature are summarised in **Table 1**. It is evident that there is a need for further research and innovation opportunities to develop the meaningful implementation of DTs in the context of BCH structural preservation. This includes addressing a systematic comparison between the concepts of DT and heritage/historic building information modelling (HBIM), defining a methodological framework for the implementation of a structural DT, developing DT as a computational model for simulations and predictive purposes, and increasing practical implementations within the structural engineering field and conservation of BCH.

**Table 1.** Application of DT paradigm for built cultural heritage (BCH) conservation.

| Application | | Type of Model |
|---|---|---|
| Graphical representation | | HBIM model |
| Information storage and data structuring | | HBIM model |
| Management planning for preventive conservation | | HBIM model |
| Digital fruition of BCH and museums | | 3D model/extended reality |
| Behaviour simulation | Environmental conditions | Numerical model |
| | Structural conditions | |
| Structural health monitoring (SHM) | | Numerical model |

DT technology is mainly used for graphical representation and storing information and data to increase knowledge and record all occurring changes, which are crucial tasks for BCH conservation. HBIM models are often chosen as digital replicas within the DT implementation to collect geometrical information and data from nondestructive tests (NDT) useful for the real-time assessment of eventual changes in structures' conditions. DT technology can also be employed for the digital fruition of BCH and museums.

However, DT technology's use for BCH conservation is not only limited to data visualisation and storage but also addresses simulation aspects, as shown by Zhang et al. [21] and Angjeliu et al. [18]. The former developed a dynamic ventilation system based on DT technology to control the relative humidity of an underground heritage site, while the latter developed a workflow for the structural assessment of the Duomo of Milan involving three steps: (i) inputs (data collection and classification); (ii) simulation model generation; and (iii) simulation model calibration and validation.

Other studies emphasise the importance of including structural health monitoring (SHM) as part of DT implementation to address the real-time connection between physical and virtual assets. Both García-Macías et al. [22] and Kita et al. [23] proposed the use of surrogate models to overcome the large computational effort required for historical structures in traditional FE-based approaches and improve the effectiveness of SHM based on automated operational modal analysis (OMA) for real-time condition assessment and continuous model updating.

Overall, the literature suggests that DT technology can be a valuable tool in conservation; even still utopian, the possibility of a full implementation breaks uncertainties typically affecting BCH structural integrity.

## 3. Not One Twin, though a Cohort of Digital Brothers to Treat Uncertainties

As a literature analysis highlights, implementing the DT paradigm for structural integrity preservation in the BCH conservation subdomain presents opportunities and challenges. One significant challenge is the domain-dependent nature of the DT concept, which means that what works effectively in other engineering-based fields may not work in BCH. While the DT paradigm was initially developed for aerospace and manufacturing domains, where there is a high level of knowledge and control over physical assets, BCH conservation involves a limited understanding of assets and handling numerous uncertainties arising from material characterisation, geometrical definition, understanding of the boundary and loading conditions, and interaction with the environment and humans [24].

In addition to these uncertainties, historical buildings were typically designed to resist gravity loads and may not have a minimum level of ductility, unlike modern buildings and industrial artefacts that are designed with ductile systems to withstand specific loading conditions. It should be noted that the term "industrial artefacts" refers to artefacts produced and assembled in a factory (i.e., belonging to the aforementioned manufacturing and aerospace domains). As a result, real-time data exchange between physical assets and their digital replicas may potentially be useless due to the fragile structural

response. This presents a significant challenge for the implementation of DT in the BCH conservation subdomain.

Developing new approaches to implementing the DT paradigm specific to the BCH conservation subdomain is essential to address this challenge. These approaches should also consider the domain-dependent nature of the DT concept and the unique characteristics of the historic structures. Moreover, it is necessary to consider how real-time data exchange can be used effectively in the context of BCH conservation, exploring different scenarios. It should be noted that historical buildings are subject to long-term damage and degradation that can occur over centuries. On the other hand, exceptional events such as earthquakes or hurricanes have a low probability of occurrence but can have severe consequences for the economy and society. In contrast, industrial artefacts are often designed using numerical optimisation tools that ensure specific performance levels even in adverse conditions.

While industrial artefacts are designed to optimise their structural performance across their life cycle, for heritage buildings, the probability of reaching a near-collapse state is unpredictable due to the nature of natural disasters, such as earthquakes and hurricanes. This unpredictability makes it challenging to evaluate the economic investment required to prevent damage and collapses of BCH in the case of a disaster. While authorities and governments invest significantly in protecting iconic buildings, this approach is not economically sustainable on a large scale, given the high number of heritage buildings in cities, particularly in Europe.

To address this gap, **Figure 5** proposes a workflow prototype concerning the implementation of the DT paradigm for the structural integrity preservation of BCH. When quantitative data, such as material properties and a full geometrical description, are difficult to detect, the definition of a data-driven DT becomes impractical. Therefore, the most rational application of the DT paradigm is to develop digital tools and numerical models that can simulate several scenarios, perform a large number of computations in a short time, and take into account the variability of some parameters. The computational results must be stored and analysed using statistical tools to develop engineering charts and decision-making tools for a rational approach. Ideally, these predictive tools must be integrated with nondestructive testing (NDT), integrating such semi-probabilistic models aiming to optimise economical resources for the structure's testing and retrofitting design.

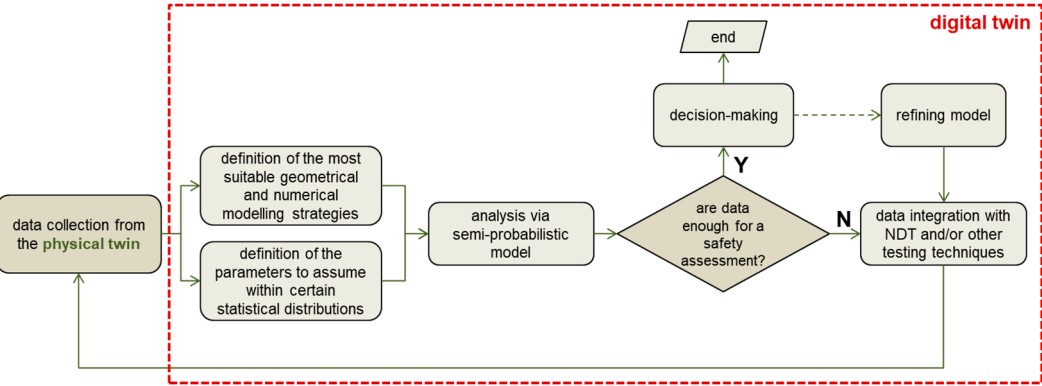

**Figure 5.** DT paradigm prototype for the structural integrity preservation of BCH.

Each iteration will allow users to work with more data and fewer uncertainties, enabling them to adopt more refined numerical models.

The following subsection discusses some insights on imperative components that must be included in a DT project: geometrical, computational modelling, and predictive tools.

### 3.1. Model Creation: Geometrical Model Generation and Computational Strategies

**Figure 6** illustrates the main approaches to generate a geometrical model from raw geometrical data. These approaches include point-cloud-based modelling, which involves processing laser scanner or photogrammetry data to produce a 3D point cloud that is then

transformed into a surface model using meshing algorithms [25–28]. Another approach is image-based modelling, which uses images captured from different perspectives to reconstruct the 3D geometry of an object. This approach is suitable for structures with complex geometries, such as decorative elements, but it requires high-resolution images and sophisticated algorithms. Other methods include manual and parametric modelling, which involve creating a geometrical model by manually inputting data or using predefined parameters, respectively. These methods are time consuming but allow for greater control over the resulting model. In addition, they allow for the use of so-defined indirect data, namely, photographs, historical drawings, and nonhomogeneous data from other domains, i.e., morphological, historical, philological, physical, and diagnostic. Due to the difficulties affecting BCH modelling, these kinds of raw data are valuable information. Finally, hybrid methods, which combine several approaches, are becoming increasingly popular, as they can leverage the advantages of different techniques while mitigating their limitations. It is worth noting that the choice of an approach depends on the available data, the level of accuracy required, and the resources available. In the case of BCH, hybrid methods, which combine automatic and manual geometrical modelling, have proven to be effective in creating accurate and detailed geometrical models [29].

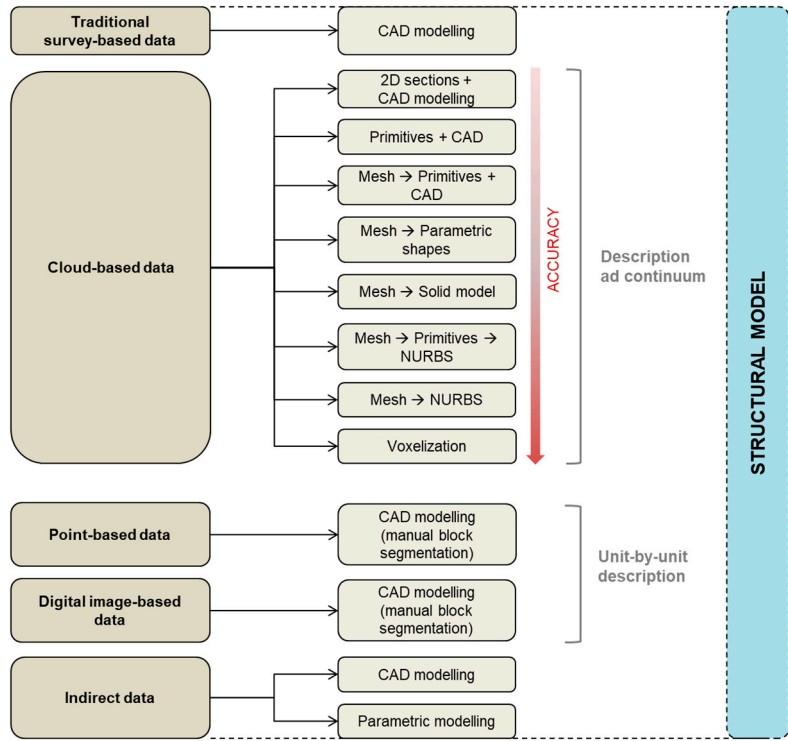

**Figure 6.** Geometrical modelling strategies classification.

The computational modelling strategy is an imperative component, and the choice of a specific strategy depends on the scenario to be simulated. The taxonomies proposed by D'Altri et al. [30], Lourenço [31], Roca et al. [32], and Saloustros [33] are taken into account, and four main approaches to the numerical modelling of masonry structures are identified. These are block-, continuum-, macroelement-, and geometry-based models.

Block-based models represent the actual masonry texture since they are modelled following the unit-by-unit representation. They are computationally demanding and more appropriate for small models and localised analysis [34–38]. Continuum models consider masonry as a deformable continuum material and are suitable for the global modelling of complex geometries involving massive elements. The most relevant issue in this approach is related to the mechanical modelling of the material properties [39–43].

In macroelement models, the structure is idealised into rigid or deformable panel-scale structural components. Macroelement models are the most widely diffused strategies for

the seismic assessment of masonry structures, especially among practitioners, due to the fact of their simplified implementation and computation. On the other hand, they present some drawbacks [44–47].

Geometry-based models assume the structure's geometry as the only input, in addition to the loading condition. The structure is modelled as a rigid body whose structural equilibrium is assessed through limit-analysis theorems, i.e., static or kinematic. Static theorem-based computational approaches can provide very useful outcomes for the investigation of the equilibrium states in masonry vaulted structures and appear especially suitable for predicting the collapse mechanism in complex masonry structures [48,49].

All of the above-discussed approaches have pros and cons. The choice to use a numerical modelling strategy rather than another depends on the type of analysis, data quality, time availability, etc. A DT, intended as a computational tool, should enable simulating different phenomena, and this will be achievable by seeking to integrate modelling strategies having a different level of complexity, from super simple and quick analytical models to super-complex multiscale or micro-modelling strategies [50].

### 3.2. Predictive Models for the Assessment of the Structural Behaviour of Heritage Buildings

One of the peculiarities of the DT paradigm is its ability to store a large amount of information and process it using high computational technology, which can help deal with uncertainties in parameters affecting the structural performance of physical assets. This feature could be worth exploring for the structural assessment of historical masonry buildings, as masonry components, such as units and joints, can exhibit wide variability due to the presence of environmental factors and the availability of raw materials during construction. Additionally, construction technology and masons' skills can influence local failure mechanisms and load-bearing capacity [24,51–54]. However, accurately predicting the structural response and addressing all types of failure modes is challenging due to the masonry's inherently nonlinear and scattered material properties [38].

Given the stochastic nature of masonry properties, probabilistic structural models that consider uncertainties and analyse their effects on the response can be reliable for addressing problems associated with random variables. Recently, some researchers have explored these models [55–57]. Furthermore, hybrid models that combine numerical and probabilistic analysis can be generated to assess the structural response of masonry structures, including the effect of uncertainties. Proper statistical or machine learning methods could improve the approach to this type of structural engineering problem, as they allow for models to be created over more complex and less understood scenarios by enriching and relaxing the sets of assumptions made on data.

### 4. Conclusions and Future Development

This paper provides an overview of the potential applications of the digital twin (DT) paradigm in the preservation of the structural integrity of built cultural heritage (BCH). However, it is worth noting that the definition and application of DT in the architecture, engineering, construction, and operation (AECO) domain are still unclear and dependent on the domain. Consequently, the full potential of the DT paradigm in the AECO domain, especially in BCH conservation, remains underutilised. Although some applications of DT have been developed, they are mostly theoretical, and few practical applications exist in the AECO domain, particularly in the context of BCH conservation.

To address these challenges and fully realise the potential of DT in the context of BCH structural preservation, the literature review presented in Section 3 provides a critical discussion and proposes a potential route for using the DT paradigm in this domain. In addition, the necessary components for applying DTs, such as generating a geometrical model, developing suitable computational modelling strategies, and implementing statistical and machine learning methods in predictive tools, are also discussed in detail.

However, to achieve the full potential of DT in BCH preservation, it is crucial to pursue further research and innovation opportunities. These include systematic comparisons be-

tween the DT and HBIM concepts to clarify their differences and interconnections, defining a methodological framework for implementing a structural DT, developing DT as a computational model for simulations and predictions, and increasing practical implementations in the field of structural engineering and BCH conservation. The correct implementation of the DT paradigm, pursued through the exploitation of its key features, would play a fundamental role not only on the issue of digital documentation of BCH, crucial within the entire conservation planning process. In fact, the use of the DT, either as a computational model or for the analysis and interpretation of data through the use of probabilistic and ML tools, would make it possible to leverage the predictive feature of the paradigm, allowing for the implementation of targeted preventive conservation strategies.

**Author Contributions:** Conceptualisation, A.V. and M.F.F.; methodology, A.V. and M.F.F.; investigation, A.V. and M.F.F.; resources, A.V. and M.F.F.; data curation, A.V. and M.F.F.; writing—original draft preparation, A.V. and M.F.F.; writing—review and editing, A.V., M.F.F. and P.B.L.; visualisation, A.V. and M.F.F.; supervision, A.V., M.F.F. and P.B.L.; project administration, P.B.L.; funding acquisition, P.B.L. All authors have read and agreed to the published version of the manuscript.

**Funding:** This research was supported by the doctoral grant PRT/BD/152822/2021 financed by the Portuguese Foundation for Science and Technology (FCT), under the MIT Portugal Program.

**Data Availability Statement:** The data supporting the reported results in the present study will be available on request from the corresponding author.

**Conflicts of Interest:** The authors declare no conflict of interest. The funders had no role in the design of the study; in the collection, analyses, or interpretation of data; in the writing of the manuscript; or in the decision to publish the results.

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
