# Peer review of "On the Use of the Digital Twin Concept for the Structural Integrity Protection of Architectural Heritage"

_infrastructures, doi:10.3390/infrastructures8050086_

Round 1
Reviewer 1 Report
1. Title: On the use of the Digital Twin concept for the structural integrity protection of Architectural Heritage
Is the paper in the focus of the journal? __x__ Yes ____ No
2. Information contained
_x_Technical note
__New techniques/theory
__New application of known concepts
__Valuable confirmation of known techniques
__Contribution contains large and realistic examples
__State-of-the-art review article
__Evaluation/validation of new systems/models
_x_The contribution is of value and interest to a significant portion of readers
__None of the above _____________
3. Conclusions drawn / outlined ideas / prototype solution
__Adequate
__Not justified
__Suffer from major omissions
__Suffer from loose generalizations
_x_Should be changed, why?
Generalized recommendations that do not take advantage of categories of digital twins that have already occurred, with an appropriate selection of their features.
4. Title
__Adequately descriptive
__Should be changed in …. Expected discussion of advantages and disadvantages of using the DT concept should be included.
5. Abstract
__Clear and adequate
_x_Should be rewritten: prediction and prevention not mentioned
__Missing
6. Keywords
__Adequately descriptive
_x_Should be changed: add preservation
7. Language
_x_Grammatically good
__Needs revision
8. Presentation and style
__Adequate
_x_Too brief for clarity
__Too comprehensive, must be shortened
__Contains irrelevant material
__Arrangement unsuitable; could be better subdivided
__Should be written and presented in professional, academic style
_x_Submission is sufficiently different from authors' previous publications.
Digital twin (DT): the paper uses the notions paradigm, model, and concept without explaining the terms.
Comprehensible assignment of features to DT’s development stages is missing
Paper is prepared according to the MDPI guidelines
___ Yes
__x_ No: title, author information, references
Does the proposed contribution contain sufficient unpublished material according to its category and text taken from the authors’ own work or that of others properly acknowledged or cited?
__x_ Yes
___ No
Why not?
9. Illustrations
__ N/A
__Number and quality adequate
__Fig(s) ...... may be omitted
__A figure is desirable to illustrate _____________________________
_x_Quality of prints/drawings inadequate
Figure 1: Quite similar to the original, poor information value. Drones are also used for data acquisition
Figure 2: Poor quality: are there copyright issues?
Figure 3: Heading on the right side: where appears the physical twin?
Figure 4: Copyright? Please add the integration of different sensor classes depending on the stages.
Deng [9] emphasizes monitoring and prediction.
Figure 5: Typo “definition” What kind of uncertainty is meant? The proposition applies to epistemic uncertainty.
Figure 6: Figure seems unbalanced and unfinished: indirect data, do you mean metadata?
By the way: there is a rich literature on geometric digital twinning of infrastructure.
10. Tables
___ N/A
_x__Not adequate Table1: format error; disappearing text
___Should be rearranged to present data more clearly
___Table ...... may be omitted
11. Abbreviations, formulae, units
__ N/A
_x_In accordance with applicable standards
__Not in accordance with applicable standards should be changed
__Should be explained
12. Literature references
__Adequate for the category of the contribution
__Inadequate
__.................. cannot be located
_x_ Following/Further references or links to organizations, software projects should be included:
1) The authors should do a literature search using the completed keywords
2) References do not match MDPI reference style requirements
3) Missing references:
AECO: https://www.digitaltwinconsortium.org/working-groups/aeco/
BIM extensions including uncertainty and sensitivity issues or other features
Banfi, F. et al. Digital Twin and Cloud BIM-XR Platform Development: From Scan-to-BIM-to-DT Process to a 4D Multi-User Live App to Improve Building Comfort, Efficiency and Costs https://www.mdpi.com/1996-1073/15/12/4497
Amirhosein Shabani et al. 3D simulation models for developing digital twins of heritage structures: challenges and strategies. Procedia Structural Integrity 37 (2022) 314-320
The authors should distinguish themselves from the following publication and highlight differences
Mahmoodian, M.; Shahrivar, F.; Setunge, S.; Mazaheri, S. Development of Digital Twin for Intelligent Maintenance of Civil Infrastructure. Sustainability 2022, 14, 8664. https://doi.org/10.3390/su14148664
Han Sun: Machine learning applications for building structural design and performance assessment: State-of-the-art review. Journal of Building Engineering Volume 33, January 2021, 101816
Yitmen, I.; Alizadehsalehi, S.; Akıner, İ.; Akıner, M.E. An Adapted Model of Cognitive Digital Twins for Building Lifecycle Management. Appl. Sci. 2021, 11, 4276. https://doi.org/10.3390/app11094276
Minor issue: [46] P. B. Lourenço, M. F. Funari, and L. C. Silva, “Building resilience and masonry structures: How can computational modelling help?,” Comput.
13. Specific and detailed comments for the authors
A revision should address the following points:
1) Figures are either adopted with little change or seem preliminary and should use previously introduced and explained terms.
2) The authors should use established terminologies in the context of DTs Maturity: partial, clone, augmented DT and their features/dimensions as well as typical realizations. Prediction and prevention issues are only succinctly addressed.
3) Prediction and prevention issues are only succinctly addressed.
4) Before using the DTs for the structural integrity protection of architectural heritage the authors should carefully introduce concepts and features of partial, clone and augmented DTs with reference to the (communication/ data transfer with the) physical twin: definitions, architecture, taxonomies, their scope of services/features and maturity, e.g., historic, heritage, geometric DT resp. content-, geometric-centric DT and their features.
5) Quality criteria and metrics are not sufficiently addressed, as well as advantages and disadvantages of using digital twins.
14. Further question
_x_ I would like to see the revision before publication.
15. Reviewer’s confidence:
___ 4 (expert)
__x_ 3 (high)
___ 2 (medium)
___ 1 (low)
___ 0 (null)
Reviewer 2 Report
The paper is well written and introduces a new perspective in the domain. It is of course susceptible of further development, which can however be the subject of future work. Anyway, it is important to publish it as is, because it is self-contained and opens new pathways to the research in the relevant domain, possibly stimulating others to contribute.
For such future work, I would suggest authors to consider DT-related work concerning cultural heritage documentation. I do not mean at all that these references should be included in the present paper, which is self-contained and does not need any expansion, not even in the section dedicated to future work. Reading such cognate work might suggest authors how to proceed in their interesting research.
In sum: publish this paper with no changes at all; invitation to authors to proceed in this interesting research; for it, dedicate some time and work to consider related topics in the field of the digital documentation of cultural heritage.
Author Response
The authors would like to thank the reviewer for the contribution and positive feedback. The authors appreciate the interesting suggestions for the future development of the work. This technical note was intended by the authors as a way of summarising what has already been discussed regarding the implementation of the DT paradigm for the civil engineering domain and to provide a concise overview, as well as a general and realistic discussion of the technological and computational tools that are considered useful for the development of the DT paradigm for the structural conservation of heritage buildings. The authors are already working on the practical application of what is anticipated in this article and on the topic of cultural heritage documentation, which they also consider to be closely linked to the concept of DTs, which will be the subject of future publications.
Reviewer 3 Report
This research deals with built cultural heritage and how DT can be an useful tool. Authors carefully explain the difference between DT and other digital tools such as BIM and CAD. Since not many papers dealing with DT and BCH are published, this research is valuable contribution.
There are some suggestions to make this article stronger.
1. Instead of discussing built cultural heritage as a general abstract concept, discussing specific examples of how DT is/can potentially be used in specific examples of BCH is necessary.
2. In section 3 (page 7), authors differentiate heritage/historical buildings and industrial artefacts. But there are overlapping cases of buildings being both industrial artefacts and heritage (i.e. industrial heritage). Author(s) should include statements that they are aware of such overlaps.
3. In section 3.1, author(s) mention that hybrid methods are better suited for BCH. But author(s) do not explain further why that is the case. Is it because hybrid method(s) can better analyze both long term damage and exceptional event (earthquake?) Please elaborate.
Minor things
- On line 139, author(s) mention figure 2.3 from another source, but it is not in the article. It is confusing because readers may think that figure 2.3 is the figure in the manuscript. Perhaps it's better not to say figure 2.3 and instead say "a figure"
Round 2
Reviewer 1 Report
Table 1 should be thoroughly revised.
- Justification leads to poor typeface
- 4D/5D simulation has not been introduced. Is BIM 4D/5D simulation - meant?
- What is meant by Brownfeld (existing) as-built survey?
Minor issue:
Table 2 :
3D model /Extendeded Reality
Figure 6 is unbalanced and looks unfinished:
Importance of indirect data for CAD/parametric modeling still not explained.
